# Phosphoribosyltransferases and Their Roles in Plant Development and Abiotic Stress Response

**DOI:** 10.3390/ijms241411828

**Published:** 2023-07-23

**Authors:** Ye Liu, Peiwen Wu, Bowen Li, Weihao Wang, Benzhong Zhu

**Affiliations:** 1College of Food Science and Nutritional Engineering, China Agricultural University, Beijing 100083, China; liuliuye7@163.com (Y.L.);; 2Key Laboratory of Plant Resources, Institute of Botany, Chinese Academy of Sciences, Beijing 100093, China

**Keywords:** phosphoribosyltransferase, nucleotides, histidine, tryptophan, NAD(P)^+^, physiological role, plant development, abiotic stress response

## Abstract

Glycosylation is a widespread glycosyl modification that regulates gene expression and metabolite bioactivity in all life processes of plants. Phosphoribosylation is a special glycosyl modification catalyzed by phosphoribosyltransferase (PRTase), which functions as a key step in the biosynthesis pathway of purine and pyrimidine nucleotides, histidine, tryptophan, and coenzyme NAD(P)^+^ to control the production of these essential metabolites. Studies in the past decades have reported that PRTases are indispensable for plant survival and thriving, whereas the complicated physiological role of PRTases in plant life and their crosstalk is not well understood. Here, we comprehensively overview and critically discuss the recent findings on PRTases, including their classification, as well as the function and crosstalk in regulating plant development, abiotic stress response, and the balance of growth and stress responses. This review aims to increase the understanding of the role of plant PRTase and also contribute to future research on the trade-off between plant growth and stress response.

## 1. Introduction

Glycosylation, the transfer of a sugar moiety to an acceptor molecule, is a widespread modification in plants that regulates gene expression and metabolite bioactivity. This glycosyl modification occurs on carbohydrates, proteins, lipids, hormones, and other secondary metabolites to alter their important properties, such as solubility, stability, biological activity, or intermolecular interactions [1,2,3,4]. Moreover, glycosylation plays a crucial physiological role in all life processes of plants, including growth, development, and stress response [2,3].

Phosphoribosylation, a special glycosyl modification catalyzed by phosphoribosyltransferases (PRTase), is a key step in the biosynthetic pathways of purine and pyrimidine nucleotides, tryptophan (Try) and histidine (His), and cofactor NAD(P)^+^ to control the production of these metabolites [5]. These metabolites are essential for plants because purine and pyrimidine nucleotides, as well as Try and His are the basic constituent units of nucleic acids and proteins, respectively. In addition, Try is also a key precursor of auxin biosynthesis [6], while NAD(P)^+^ is the core substance of energy metabolism [7].

The physiological functions of many plant PRTases have been clarified [8,9,10,11,12]. In addition, PRTases can collectively contribute to plant life processes since different PRTases crosstalk with each other by sharing 5-phosphoribosyl-1-pyrophosphate (PRPP) as sugar donors. However, research on plant PRTases in the past few decades mainly focuses on their role in plant survival and thriving, with little discussion about their role and crosstalk in plant development and abiotic stress responses [13,14]. In fact, recent studies clearly demonstrate that PRTase-related metabolic pathways are critical for chloroplast development, gametophyte development, salt, and osmotic stress response [15,16,17,18]. Herein, we summarize the recent advances in the function and crosstalk of plant PRTases in regulating plant development and abiotic stress response, aiming to provide new insights into the complicated role of PRTase in plant life processes.

## 2. Classification and Characteristics of PRTase

To date, studies have shown that PRTase is responsible for catalyzing the transfer of ribose-5-phosphate from the glycosyl donor PRPP to acceptor molecules (e.g., adenine, guanine, uracil) to form glycosidic bonds, which rely on divalent cations [5,19]. PRTase belongs to the PRT family and is identified at the amino acid sequence level [5]. Almost all PRTases contain a 13-residue sequence motif that is predicted to be a PRPP-binding site, which is composed of four hydrophobic amino acids, two acidic amino acids and seven variable characteristic amino acids [5,20].

PRTases can be divided into four categories based on the similarity of the three-dimensional structures, as shown in Figure 1A [5,19]. Class I PRTase is a homodimer formed by the *N*-terminal domain of one monomer adjoining the *C*-terminal domain of the other monomer, which shares a common α/β-barrel domain [5]. Class II PRTase monomers have two α/β-barrel domains that form homodimers in a similar manner to class I PRTase [21]. Class III PRTase monomers contain a small *N*-terminal α-helical domain and a large *C*-terminal α/β-barrel domain, with the *N*-terminal domain of one monomer close to the *N*-terminal domain of the other monomer, forming homodimers [19,22,23]. Class IV PRTase are homohexamers, assembled from six monomers, each containing three α/β barrel domains [24,25].

As shown in Figure 1B and Table 1, different classes of PRTase are involved in different metabolic pathways. Class I PRTase is responsible for purine and pyrimidine nucleotide biosynthesis pathways, including amidophosphoribosyltransferase (ATase), adenine phosphoribosyltransferase (APRT), hypoxanthine-guanine phosphoribosyltransferase (HGPRT), orotate phosphoribosyltransferase (OPRT), and uracil phosphoribosyltransferase (UPRT) [5,26,27]. Class II PRTase is involved in the NAD(P)^+^ biosynthesis pathway, including quinolinic acid phosphoribosyltransferase (QPRT) and nicotinamide phosphoribosyltransferase (NaPRT) [21,28]. Class III PRTase participates in the Try biosynthesis pathway, with only anthranilate phosphoribosyltransferase (AnPRT) [19,22,23,29]. Class IV PRTase is responsible for the His biosynthesis pathway, with only ATP phosphoribosyltransferase (ATP–PRT) [24,25].

**Figure 1 ijms-24-11828-f001:**
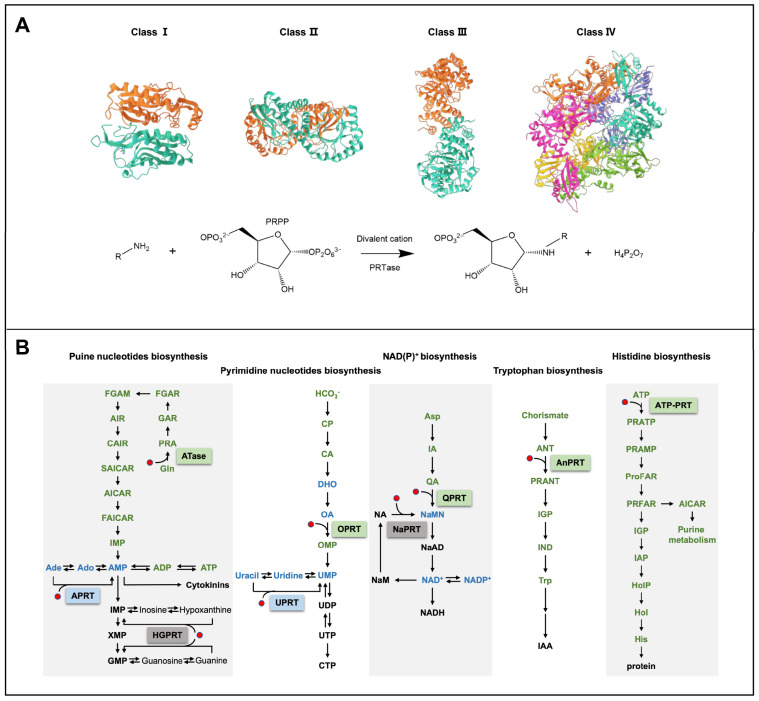
The protein structure and the catalytic reaction of PRTase, and its related metabolic pathways in plants [30,31,32,33]. (**A**) Protein structure and the catalytic reaction of PRTase. Different colors represent different monomers in the homodimer or homomultimer. (**B**) The metabolic pathways that PRTases are involved in. PRTases are shown within boxes. The green box indicates that it is chloroplast-localized, the blue box indicates that it can be chloroplast- and cytoplasm-localized, and the gray box indicates that it is cytoplasm-localized. The red dot represents 5-phosphoribosyl-1-pyrophosphate (PRPP), the common substrate of PRTases. Abbreviations for metabolites are shown in the Abbreviations list.

## 3. Functions of PRTase and Related Metabolic Pathways in Plants 

### 3.1. Class I PRTase

Class I PRTase, which comprises more family members compared with the other class of PRTase, is responsible for the biosynthesis of all nucleotides, including adenine, guanine, cytosine, and uracil nucleotides. As is known, nucleotides are essential for all organisms because they are the fundamental building blocks of nucleic acids as well as ribosomes. Furthermore, adenosine triphosphate (ATP) is required for the biosynthesis of all other nucleotides, and adenine nucleotides have a greater effect on plant physiology than other nucleotides [30]. Thus, this section will focus on the role of adenine nucleotide biosynthesis in plant growth and development.

#### 3.1.1. The Role of Class I PRTase in the Adenine Nucleotide Biosynthesis Pathway

In class I PRTase, only ATase and APRT are involved in adenine nucleotide biosynthesis. ATase and APRT are key factors that promote adenine nucleotide accumulation in plants [41]. ATase catalyzes the first and rate-limiting step in the adenine nucleotide de novo biosynthesis pathway [42]. In Arabidopsis (*Arabidopsis thaliana*), three *ATase* genes have been identified. *AtATase1* is specifically expressed in flowers and roots, and *AtATase2* is rather constitutively expressed in leaves, flowers, and roots, while *AtATase3* is weakly expressed in silique, cauline leaves, and roots [8,42]. APRT is a key enzyme in the adenine salvage pathway that converts adenine to adenosine monophosphate (AMP) [13]. Three *APRT* genes have been identified in Arabidopsis, with AtAPRT1 contributing most of the APRT activity, as the APRT activity of the *AtAPRT1* knockout mutant was reduced by >95% [13,43]. In addition to Arabidopsis, *APRT* genes have been found in barley, rice, wheat, oilseed rape, maize, and tomato [44].

#### 3.1.2. Adenine Nucleotide Regulates Plant Growth and Development

Altered adenine nucleotide biosynthesis causes changes in the intracellular level of adenine nucleotides, thus regulating plant growth and development. Reduced biosynthesis of adenine nucleotides, especially ATP, strongly inhibits RNA synthesis in plants because ATP is the basis for the biosynthesis of all other nucleotides. Notably, phosphate is critical for the biosynthesis of ATP [30]. In Arabidopsis, phosphate starvation leads to a significant reduction in intracellular ATP levels and a nearly 90% reduction in the RNA content of shoots, resulting in a growth arrest in seedlings [45]. In addition, transcriptional expression of the *APRT* gene in the adenine nucleotide biosynthesis pathway was significantly induced. A similar growth arrest phenotype also appears in the suspension-cultured *Catharanthus roseus* cells under phosphate starvation conditions [46]. The addition of phosphate could rapidly increase the RNA content in the *Catharanthus roseus* cells within 24 h, and the cells would be able to grow again [46]. Thus, adenine nucleotide is a key factor that regulates RNA synthesis and plant growth.

In addition to reducing RNA synthesis, defects in adenine nucleotide biosynthesis also inhibit ribosome biogenesis in plants. Adenine nucleotide starvation caused by ATase deficiency in the de novo biosynthesis pathway can activate autophagy and inhibit the evolutionarily conserved TOR (target of rapamycin) kinase activity, a core regulator of ribosome biogenesis and plant growth [47]. In addition, silencing the expression of the PRPP synthetase (*PRS4*) in *N. benthamiana* reduces the adenine nucleotide biosynthesis and thus inhibits TOR activity and ribosome biogenesis, resulting in pleiotropic phenotypes including dwarfism, abnormal leaf shape, and delayed flowering [48]. Hence, adenine nucleotide contributes to plant growth and development by regulating TOR activity and ribosome biogenesis (Figure 2).

As derivatives of adenine, the synthesis of cytokinins also requires class I PRTase [49,50]. Notably, ATase is critical for cytokinin synthesis. Arabidopsis *ATase2* mutants have shown decreased cytokinin content, cell number, and the level of plastid-encoded RNA polymerase (PEP)-dependent transcript in leave cells, resulting in impaired chloroplast development, growth retardation, and bleached/etiolated seedling phenotype, and similar phenotypes have also been found in transgenic tobacco plants [8,11,15,34,35]. In addition, APRT participates in cytokinin inactivation. Arabidopsis *APRT1* mutants have elevated levels of activated cytokinin, which triggers cytokinin responses, resulting in increased chlorophyll and anthocyanin contents, and delayed leaf senescence [13,14]. Furthermore, APRT also plays an important role in gametophyte development in Arabidopsis, wheat, rice and *Vitis amurensis* [16,36,37,51]. Overall, ATase and APRT play a key role in plant chloroplast and gametophyte development by regulating the accumulation of adenine nucleotides and cytokinins (Figure 2).

### 3.2. Class II PRTase

#### 3.2.1. The Role of Class II PRTase in the NAD(P)^+^ Biosynthesis Pathway

QPRT and NaPRT, the only two members of class II PRTase, are rate-limiting enzymes in the biosynthesis pathways of NAD(P)^+^ [7,9]. QPRT is responsible for the de novo biosynthesis pathway of NAD(P)^+^, while NaPRT is involved in the salvage pathway of NAD(P)^+^. Class II PRTase promotes the intracellular accumulation of NAD(P)^+^ and is essential for maintaining NAD(P)^+^ homeostasis in plant cells since the half-life of NAD(P)^+^ can be as short as 15 min [52,53].

#### 3.2.2. NAD(P)^+^ Regulates Plant Growth and Development

The change of intracellular NAD(P)^+^ level strongly affects plant growth and development. NAD^+^, an indispensable coenzyme and redox carrier in plants, is involved in many redox reactions in cellular glycolysis, the tricarboxylic acid cycle, and other energy metabolic processes. In these processes, NAD^+^ is reduced to NADH, which is the basis for mitochondrial respiration to generate ATP through oxidative phosphorylation. Decreased free NAD^+^ levels severely affect mitochondrial-related metabolic processes, thereby greatly inhibiting ATP biosynthesis and cellular energy metabolism [7]. Furthermore, enzymes in the NAD^+^ biosynthesis pathway are essential for the plant, such that loss-of-function mutants are embryo-lethal [9].

In addition to its role in energy metabolism, NAD(P)^+^ also contributes to chloroplast development and photosynthesis. Recent studies have reported that NAD^+^ cap modification at the 5′ end of mRNA can regulate mRNA stability and translation in a DOX1-dependent manner [54,55,56,57]. Arabidopsis *Atdxo1* mutant has pleiotropic phenotypes, such as dwarfism, leaf yellowing/albinism, and various developmental defects [17]. This is due to the increased NAD^+^-RNA stability in the mutant *Atdxo1*, and thus, enhances post-transcriptional silencing of specific transcripts mainly involved in porphyrin and chlorophyll metabolism, as well as photosynthesis [17,58]. Additionally, NADP^+^ is an indispensable coenzyme in plant photosynthesis [7,59], because NADP^+^ is reduced in the photoreaction to NADPH that is required for the CO_2_ fixation in the dark reaction. Silencing of the *QPRT* gene reduces the content of NAD(P)^+^, leading to a decrease in chlorophyll content, inhibition of photosynthesis, and delayed growth and development in *Nicotiana tabacum* [39]. Additionally, mutation of rice *OsNaPRT1* results in dwarfism and early leaf senescence [12]. These results indicate that QPRT and NaPRT are important contributors to plant growth and development by maintaining moderate intracellular NAD(P)^+^ levels (Figure 3).

#### 3.2.3. NAD(P)^+^ Regulates Plant Abiotic Stress Response

Recent studies have found that the crosstalk between NAD(P)^+^ and abscisic acid (ABA) is involved in regulating plant abiotic stress responses. In plants, ABA is the most important hormone in response to abiotic stress, with functions such as stress resistance, growth inhibition, seed germination inhibition, and senescence promotion [60]. Interestingly, high levels of NAD(P)^+^ can promote ABA synthesis and signal transduction, as well as enhance plant response to ABA and stress. In turn, ABA signaling can prevent the further increase in NAD(P)^+^ levels through the downstream transcription factor ABI4 specifically inhibiting the expression of quinolinate synthase gene in the NAD(P)^+^ de novo biosynthesis pathway [60].

High levels of NAD(P)^+^ contribute to plant abiotic stress response by promoting ABA synthesis and signal transduction (Figure 3). ABA levels increase in an NAD^+^-dependent manner in seedlings grown on mediums with different concentrations of NAD^+^ [60,61,62]. Under abiotic stress conditions, ABA activates reactive oxygen species (ROS) signaling; in turn, ROS can inactivate tryptophan synthase TSB1 through sulfinylation, thereby relieving its inhibition on the activity of β-glucosidase BG1 and promoting the release of ABA from ABA-GE to form a positive feedback loop [63]. Importantly, ABA-activated ROS signaling, which is critical for plants to respond to salt, drought, high light and other stresses, is mainly produced by NADPH oxidase (RBOHF and RBOHD) [60,64,65,66]. In fact, RBOHF and RBOHD require NADPH electrons to reduce oxygen to superoxide anions, so the continuous supply of NADPH is a necessary fuel to maintain ROS production [67]. Defects in NAD(P)^+^ biosynthesis lead to RBOHF and RBOHD inactivation, resulting in the inhibition of ROS production. NADP^+^ and ROS levels have been observed to be reduced in the *Atfin4* and *Atfin4-4* mutants with impaired NAD(P)^+^ de novo biosynthesis [67]. Thus, high levels of NAD(P)^+^ promote ABA synthesis and ABA-activated ROS signaling under abiotic stress conditions.

In addition to promoting ABA synthesis and signaling, high levels of NAD^+^ enhance plant responses to ABA and abiotic stress by activating NAD^+^-dependent enzymes (Figure 3). NAD^+^ acts as a coenzyme involved in ADP-ribosyl cyclase-mediated calcium-responsive second messenger cyclic ADP-ribose (cADPR) synthesis [68], sirtuins-mediated protein deacetylation [69], and poly (ADP-ribose) polymerase (PARP)-mediated protein single- and poly-ADP ribosylation [70]. These NAD^+^-dependent enzymes consume NAD^+^ and are sensitive to free NAD^+^ levels. Elevating NAD^+^ levels by silencing the expression of *AtPARP2* could promote ADP-ribosyl cyclase-mediated cADPR synthesis to activate Ca^2+^ signaling, thereby enhancing plant response to ABA [71]. cADPR is a key regulator of the ABA signaling pathway in plants and induces the expression of more than 100 ABA-responsive genes which subsequently enhance the activity of ADP-ribosyl cyclase, further amplifying cADPR signaling [72]. In addition, the NAD^+^-dependent histone, H3K9 deacetylase SRT1, also positively regulates the expression of ABA-responsive genes and plant stress responses. For example, the Arabidopsis SRT1 mutant, *Atsrt1*, and its RNAi-silenced lines have exhibited decreased sensitivity to ABA and stress, while the overexpressing lines have exhibited hypersensitivity to salt and osmotic stress due to their excessive response to ABA [18]. Similarly, *Oryza sativa* (rice) SRT1 is also involved in the epigenetic regulation of stress-responsive genes [73,74,75]. Altogether, high levels of NAD^+^-induced calcium signaling and histone deacetylation greatly enhance plant responses to ABA and abiotic stress.

Interestingly, NAD^+^ can also improve plant stress resistance. Under abiotic stress conditions, NAD^+^ is mainly consumed by PARP-mediated modification of nuclear proteins, thereby regulating DNA synthesis and repair, chromatin structure, gene transcription, and cell cycle [7]. This PARP-mediated modification is one of the major causes of energy depletion and death in mammalian cells [71,76]. In Arabidopsis and *Brassica napus*, RNAi silencing of *PARP* expression has been found to reduce NAD^+^ depletion resulted in broad-spectrum stress-resistant plants without negative effects on plant growth, development, and seed production [71]. In addition, overexpression of the ADP-ribose pyrophosphatase gene *NUDX2*, which recycles AMP and ribose-5-phosphate from free ADP-ribose molecules has been found to maintain ATP and NAD^+^ levels and enhance tolerance to oxidative stress in Arabidopsis [77]. Thus, maintaining NAD^+^ levels under abiotic stress can greatly improve plant stress resistance and thus regulate plant growth-defense balance.

### 3.3. Class III PRTase

#### 3.3.1. The Role of Class III PRTase in the Tryptophan Biosynthesis Pathway

AnPRT is the only member of class III PRTase and catalyzes the second step of Try biosynthesis, which transfers the phosphoribosyl group of PRPP to anthranilate to form phosphoribosyl anthranilate [23]. Overexpression of PRPP synthetase gene to increase PRPP content in Arabidopsis can promote Try biosynthesis, indicating that this step is the key to determining the flow of this pathway [78]. Thus, AnPRT is a critical factor regulating Try accumulation.

#### 3.3.2. Tryptophan Regulates Plant Growth and Development

As the primary precursor of IAA synthesis, the level of Try is an important determinant of IAA synthesis [6,32,79]. Changes in Try biosynthesis affect its intracellular levels and thus regulate the synthesis of IAA, a major plant hormone that regulates plant growth [80]. In the *Attsb1* mutant whose mutation occurred in the tryptophan synthase β-subunit (TSB) loci, both IAA biosynthesis and plant growth are severely inhibited [63]. Furthermore, the Try biosynthesis-deficient mutant, *tdd1*, is embryonic-lethal in rice [81]. On the contrary, transgenic *Brassica oleracea* overexpressing *BoTSB1* or *BoTSB2* can accumulate more Try and IAA, so it exhibits phenotypes such as long hypocotyls, large plants, and a high number of lateral roots [82]. Thus, Try biosynthesis is essential for maintaining moderate IAA levels and plant growth.

Moderate levels of IAA in the cytoplasm and apoplasts can jointly promote plant growth and development through signal transduction. On the one hand, cytoplasmic IAA is recognized by the receptor TIR1 and promotes the degradation of canonical AUX/IAA inhibitors, thereby releasing auxin response factors (ARFs) to promote plant growth [83]. In addition, *Solanum lycopersicum* (tomato) ARFs have been found to be involved in chloroplast development, since SlARF4, SlARF10, and SlARF6A promote chlorophyll synthesis, photosynthesis, and increase starch accumulation [84,85,86]. On the other hand, unlike TIR1-dependent cytoplasmic signaling, moderate levels of IAA in the apoplast activate the small G protein ROP2 (Rho-related protein in plant 2) via transmembrane kinase 1 (TMK1), thereby activating TOR to promote plant growth [87,88]. Furthermore, TOR phosphorylates and stabilizes the IAA efflux carrier, pin-formed2 (PIN2), and its gradient distribution in Arabidopsis taproots, stimulating root tip elongation [89]. Overall, moderate levels of IAA promote plant growth and development by activating ARFs and TOR (Figure 4).

#### 3.3.3. Tryptophan Regulates Plant Abiotic Stress Response

Try contributes to plant abiotic stress responses by promoting the accumulation of high levels of IAA. In Arabidopsis, loss of function of anthranilate synthase α and β subunits (WEI2/ASA1 and WEI7/ASB1) can block excessive IAA accumulation and corresponding phenotypes, such as growth inhibition of seedling roots, epinasty of the cotyledons, and adventitious root formation [90]. Moreover, these phenotypes can be complemented by supplementing Try in the medium [90]. Hence, excess biosynthesis of Try is essential for the accumulation of high levels of IAA.

High levels of IAA inhibit IAA signaling and activate ABA signaling to prompt plant abiotic stress responses and inhibit plant growth (Figure 4). In Arabidopsis, high levels of IAA in the apoplast binding to the receptor ABP1 trigger the *C*-terminal cleavage of TMK1 (TMK1-C) [91]. TMK1-C subsequently enters the nucleus to activate non-canonical IAA32 and IAA34, thus repressing the expression of ARFs [83]. Moreover, TMK1-C also facilitates the phosphorylation and inactivation of the ABA signaling negative regulator PP2Cs (protein phosphatase 2C, ABI1/2), which releases its interacting proteins SnRK2s, thereby activating SnRK1 to inhibit TOR [92,93]. Notably, the inhibition of plant growth by ABA signaling is largely dependent on the evolutionarily conserved energy sensor, SnRK1, and its inhibition of TOR. Under normal conditions (nutrition adequacy), ABA signaling and SnRK1 signaling are blocked, thereby maintaining TOR activity and promoting plant growth, whereas under nutrient limitation conditions, ABA signaling and SnRK1 are activated, and activated SnRK1 inhibits TOR activity and plant growth [92,94]. Specifically, SnRK1 activates specific transcription factors (such as bZIP63, WRKYs, and NACs) and epigenetic factors (such as H3K27me3 demethylase JMJ705) [95,96,97,98,99], and inactivates TOR to reduce global H3K27me3 and DNA methylation, as well as allows EIN2 to negatively regulate the expression of genes related to plant cell division and elongation [100,101,102,103,104], thereby promoting catabolism and inhibiting anabolism to maintain cellular energy homeostasis under stress. Taken together, the level of IAA is a decisive factor in the trade-off between plant growth and stress response.

### 3.4. Class IV PRTase

#### 3.4.1. The Role of Class IV PRTase in the Histidine Biosynthesis Pathway

ATP–PRT, the only member of class IV PRTase, catalyzes the first step of His biosynthesis, which is the condensation of ATP and PRPP to form phosphoribosyl ATP (PRATP) [40]. ATP–PRT is the rate-limiting enzyme in the His biosynthesis pathway and is feedback inhibited by His [105]. ATP–PRT is a key regulator of His accumulation since the level of His in plants largely depends on ATP–PRT activity [33]. There are two ATP–PRT homologous proteins in Arabidopsis with 74.6% similarity, and both contain chloroplast transit peptides at the *N*-terminus [106]. Overexpression of either isoform *ATP–PRT* is sufficient to increase free His level by 41-fold in Arabidopsis bud tissue without altering the content of other amino acids [33].

#### 3.4.2. Histidine Regulates Plant Growth and Development

His is an essential amino acid for plant growth and development. In Arabidopsis, mutations at the His biosynthesis pathway gene are lethal [33]. His residue functions as a conserved catalytic site for many enzymes due to its imidazole functional group, which gives it a unique advantage in participating in acid-base catalyzed reactions [33]. Among these enzymes, histidine kinases such as cytokinin receptors, ethylene receptors, and phytochromes are widely involved in signal transduction during plant growth and development [107]. The biosynthesis of His appears to be a limiting factor for the biosynthesis of these important histidine kinases since His is one of the least abundant amino acid residues in proteins [33]. Thus, changes in His biosynthesis may influence cytokinin receptor synthesis to regulate cytokinin signaling and chloroplast development (Figure 5).

His biosynthesis may also affect cellular energy homeostasis (Figure 5). The biosynthesis of His is an energy-consuming process, since it is estimated that 31–41 ATPs are required for one His molecule, and cells with uncontrolled His biosynthesis consume 2.5% of its total metabolic energy [33,108,109]. Furthermore, the His biosynthesis pathway also releases the adenine nucleotide intermediate AlCAR into the de novo adenine nucleotide biosynthesis pathway to generate AMP [33], thereby downregulating the intracellular energy charge. Importantly, a significant negative correlation between biomass and His content has been observed in Arabidopsis overexpressing *AtATP–PRT* [33]. These results suggest that moderate His biosynthesis is essential to maintain energy homeostasis and plant growth.

## 4. Crosstalk between the PRTase-Related Metabolic Pathways

Due to sharing the substrate PRPP, PRTase leads to crosstalk between related metabolic pathways. Furthermore, PRTase-related metabolic pathways play a synergistic role in regulating plant growth, development, and abiotic stress response.

### 4.1. Common Substrate of PRTases

PRPP, the intersection of PRTase-related metabolic pathways, is essential for all organisms. The competitive utilization of the common substrate PRPP by different PRTase makes it flow to the corresponding metabolic pathway. Hove-Jensen et al. [110] reported that PRPP mainly flows to the purine and pyrimidine nucleotide biosynthesis pathways. In *Escherichia coli*, purine and pyrimidine nucleotide biosynthesis each consume 30% to 40% of PRPP, His and Try biosynthesis each consume 10% to 15% of PRPP, and NAD(P)^+^ biosynthesis consumes approximately 1% of PRPP [110]. However, the competition and distribution of PRPP in PRTase-related metabolic pathways in plants remain to be elucidated. Notably, the content of PRPP is an important factor limiting plant growth since overexpression of the PRPP synthetase gene significantly increases biomass accumulation in Arabidopsis and *Nicotiana tabacum* [78]. Hence, the competitive utilization of limited PRPP by PRTase regulates the biosynthesis of PRTase-related metabolites during plant growth.

### 4.2. Common Function of PRTase-Related Metabolic Pathways

The PRTase-related metabolic pathways coordinately promote plant growth and development (Figure 6A). Adenine nucleotide and His, Try, NAD(P)^+^ promote cytokinin signaling, IAA signaling, and mRNA 5′ NAD^+^ cap modification, respectively, and thus participate in chloroplast development and photosynthesis. Moreover, photosynthesis-derived glucose generates ATP through oxidative phosphorylation, which together with moderate levels of IAA activates TOR [111]. Importantly, TOR promotes central carbon and energy metabolism, as well as key anabolic processes, including nucleotide, amino acid, lipid, and cell wall synthesis, which are essential for rapid growth [88,100,101,111,112,113,114,115]. Overall, the PRTase-related metabolic pathways are jointly involved in plant growth and development by promoting chloroplast development and activating TOR.

PRTase-related metabolic pathways also synergistically affect plant abiotic stress responses and the switching between growth and stress responses (Figure 6B). Under abiotic stress conditions, high levels of NAD^+^ and IAA directly activates ABA signaling in response to nutrient deficiencies and energy limitations. Importantly, changes in the flow of any PRTase-related metabolic pathway may alter the flux of PRPP to NAD^+^ or IAA biosynthesis pathways, leading to plant abiotic stress responses. For example, although the NAD^+^ content of multiple NAD^+^ biosynthesis-deficient mutants in Arabidopsis and rice is significantly reduced, they instead have an ABA hypersensitivity phenotype and excessive accumulation of ROS, resulting in growth inhibition and premature leaf senescence [12,60,116,117]. In addition, mutations in the adenine nucleotide salvage enzyme AtAPRT1 lead to enhanced oxidation and high-temperature tolerance in Arabidopsis [13]. His biosynthesis is also a potentially important factor in inhibiting plant growth under energy-limited conditions [33].

In a mechanism similar to that of promoting plant abiotic stress response, PRTases may participate in fruit ripening (Figure 7). To date, significant increases in free NAD^+^ and IAA levels have been observed before the initiation of tomato fruit ripening (i.e., system II ethylene production) [31,118]. High levels of NAD^+^ and IAA may be critical for system II ethylene production and fruit ripening since they can trigger ABA signaling that mediates SlSnRK1 activation. On the one hand, activated SlSnRK1 may promote the expression of the key regulator *SlRIN* in the autocatalytic loop of system II ethylene and the ethylene biosynthesis genes by inactivating SlTOR and activating demethylase to remove H3K27me3 and DNA methylation [99,119]. In fact, heterologous overexpression of *Malus hupehensis SnRK1* advanced tomato fruit ripening by about 10 days [120,121], whereas the silencing of *SlSnRK1* expression in tomato fruit by VIGS delayed or completely inhibited ripening [122]. On the other hand, in addition to increased ethylene biosynthesis, activated SlSnRK1 may also enhance fruit sensitivity to ethylene, which is necessary for fruit ripening [123,124]. Specifically, SlSnRK1-mediated SlTOR inactivation stabilizes SlEIN3/EIL1 in a SlEIN2-dependent manner to induce downstream ethylene-responsive factors expression, resulting in increased fruit sensitivity to ethylene [101,125]. Huang et al. [126] constructed the tomato SlEIN2 mutant *slein2-1* and found it was insensitive to ethylene and completely arrested fruit ripening. Taken together, PRTase-related metabolic pathways are potentially critical in fruit ripening, mediating SlSnRK1 activation and SlTOR inactivation through the activation of ABA signaling.

## 5. Conclusions and Perspectives

PRTases contribute to nucleic acid and protein synthesis, energy metabolism, hormonal signaling, and epigenetics by promoting the biosynthesis of adenine nucleotides, NAD(P)^+^, Try, and His. Importantly, these metabolites play an essential role in chloroplast development and plant growth through cytokinin signaling, IAA signaling, and NAD^+^ cap modification of mRNA. In addition, PRTases co-regulate plant abiotic stress response via crosstalk with each other. Under abiotic stress conditions, increased biosynthesis of NAD^+^ and Try leads to elevated levels of free NAD^+^ and IAA, which govern the transition of plants from growth to stress response through ABA signaling and epigenetic remodeling. In this process, NAD-dependent enzymes and the conserved SnRK1-TOR axis regulatory modules are key factors, down-regulating anabolism and up-regulating catabolism through histone acetylation/methylation remodeling and transcriptional reprogramming to coordinate nutrient supply and plant growth. 

It is a brilliant strategy for plants to use the levels of PRTase-related metabolites that are necessary for growth as a signal to guide the transition between plant growth and stress response. Specifically, plants can sense a stressful environment by monitoring elevated levels of these intracellular metabolites, rapidly turning on the switch of stress response, which then triggers a reduction in these metabolite levels, further turning off the switch of growth, and vice versa. Until now, studies have found that levels of NAD^+^ and Try play a decisive role in the trade-off between plant growth and stress response, but the importance of adenine nucleotides and His has been underestimated. Elucidating the role and mechanism of adenine nucleotide and His biosynthesis in plant growth defense tradeoffs is an urgent but challenging goal. This knowledge will guide us in precisely regulating the levels of PRTase-related metabolites in plants to help maximize plant yield under fluctuating environmental conditions.

## Figures and Tables

**Figure 2 ijms-24-11828-f002:**
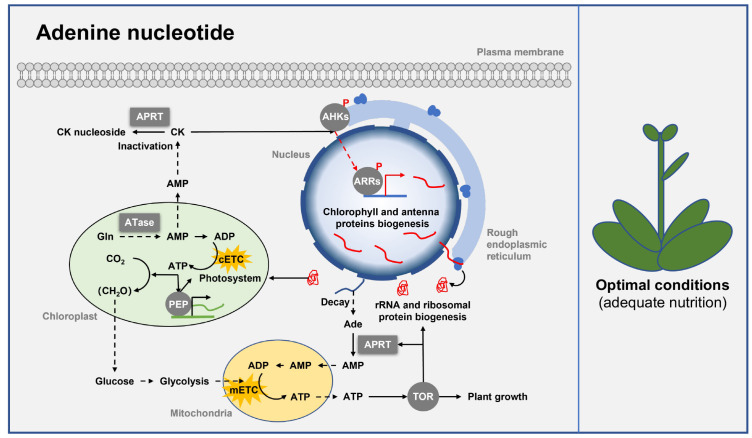
Class Ⅰ PRTase in adenine nucleotide biosynthesis pathways regulates plant growth and development. Class Ⅰ PRTase ATase and APRT regulate chloroplast development, ribosome biogenesis, and plant growth under nutrient-adequate conditions by influencing the intracellular levels of adenine nucleotides and CK. ATase and APRT are represented by square boxes, and other proteins are represented by round boxes. Solid arrows indicate direct interactions or one-step reactions, and dashed arrows indicate indirect interactions, multistep reactions, or transmembrane transport. Abbreviations for metabolites are shown in the Abbreviations list. Other abbreviations are as follows: adenine phosphoribosyltransferase (APRT), amidophosphoribosyl transferase (ATase), plastid-encoded RNA polymerase (PEP), Arabidopsis histidine kinase (AHK), Arabidopsis response regulator (ARR), target of rapamycin (TOR), chloroplastic electron transport chain (cETC), mitochondrial electron transport chain (mETC).

**Figure 3 ijms-24-11828-f003:**
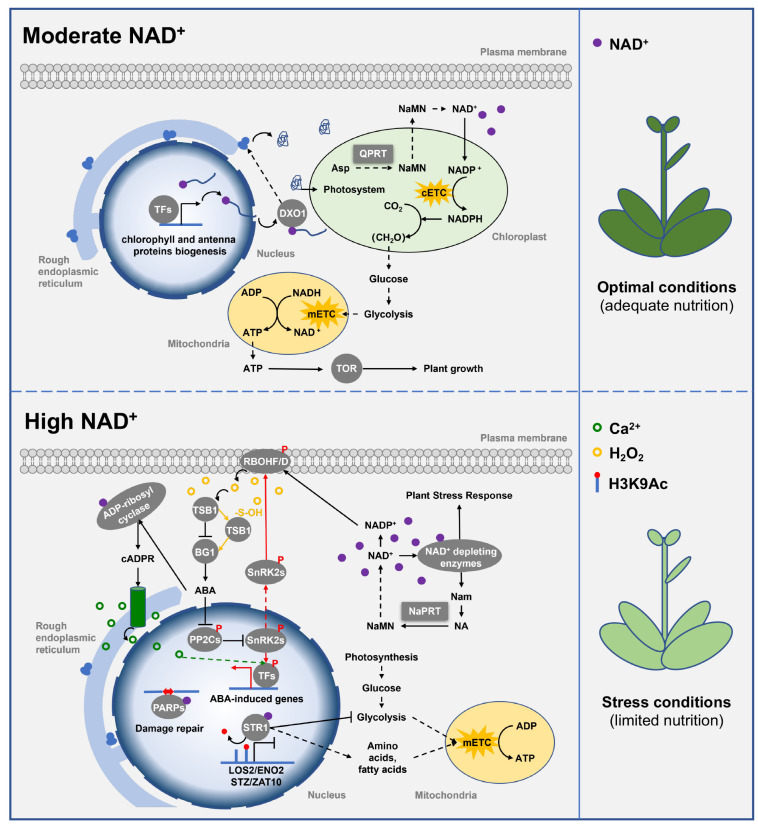
Class II PRTase in NAD(P)^+^ biosynthesis pathways regulate plant growth, development, and abiotic stress response. Class II PRTase QPRT and NaPRT control the biosynthesis of NAD(P)^+^ to regulate its intracellular levels. Under nutrient-adequate conditions, moderate levels of NAD^+^ participate in mRNA modification and ATP biosynthesis to promote chloroplast development and plant growth. However, under nutrient-limiting conditions, elevated levels of NAD^+^ enable NAD^+^-dependent enzymes to trigger calcium signaling and histone deacetylation, thereby activating ABA signaling, and ROS signaling to promote plant systemic stress responses. QPRT and NaPRT are represented by square boxes, and other proteins are represented by round boxes. Solid arrows indicate direct interactions or one-step reactions, and dashed arrows indicate indirect interactions, multistep reactions, or transmembrane transport. Abbreviations for proteins are as follows: quinolinic acid phosphoribosyltransferase (QPRT), decapping exonuclease 1 (DXO1), transcription factors (TFs), target of rapamycin (TOR), NADPH oxidase (RBOHF/D), tryptophan synthase β subunit 1 (TSB1), β-glucosidase 1 (BG1), sucrose non-fermenting 1-related protein kinase 2 (SnRK2), nicotinamide phosphoribosyltransferase (NaPRT), protein phosphatase 2C (PP2C). Abbreviations for metabolites are shown in the Abbreviations list.

**Figure 4 ijms-24-11828-f004:**
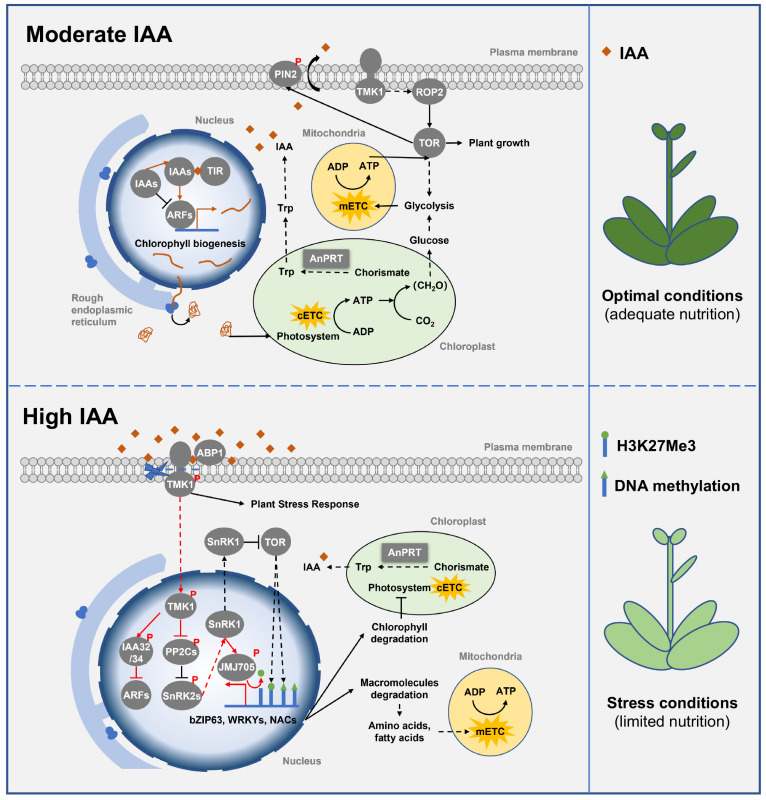
Class Ⅲ PRTase in the tryptophan biosynthesis pathway regulates plant growth, development, and abiotic stress response. Class Ⅲ PRTase AnPRT controls the biosynthesis of Try to regulate the level of IAA. Under nutrient-adequate conditions, moderate levels of IAA promote ARFs expression and TMK1-mediated TOR activation to promote chloroplast development and plant growth. However, under nutrient-limiting conditions, elevated levels of IAA mediate C-terminal cleavage of TMK1 to repress the expression of ARFs and activate ABA signaling, thus activating SnRK1 and inactivating TOR to promote plant abiotic stress responses. AnPRT is represented by a square box, and other proteins are represented by round boxes. Solid arrows indicate direct interactions or one-step reactions, and dashed arrows indicate indirect interactions, multistep reactions, or transmembrane transport. Abbreviations for proteins are as follows: pin-formed 2 (PIN2), transmembrane kinase 1 (TMK1), rho-related protein in plant 2 (ROP2), target of rapamycin (TOR), indole-3-acetic acid (IAA), Toll–interleukin 1 receptor (TIR), auxin response factor (ARF), anthranilate phosphoribosyltransferase (AnPRT), auxin-binding protein 1 (ABP1), sucrose non-fermenting 1-related protein kinase 1 (SnRK1), protein phosphatase 2C (PP2C), sucrose non-fermenting 1-related protein kinase 2 (SnRK2). Abbreviations for metabolites are shown in the Abbreviations list.

**Figure 5 ijms-24-11828-f005:**
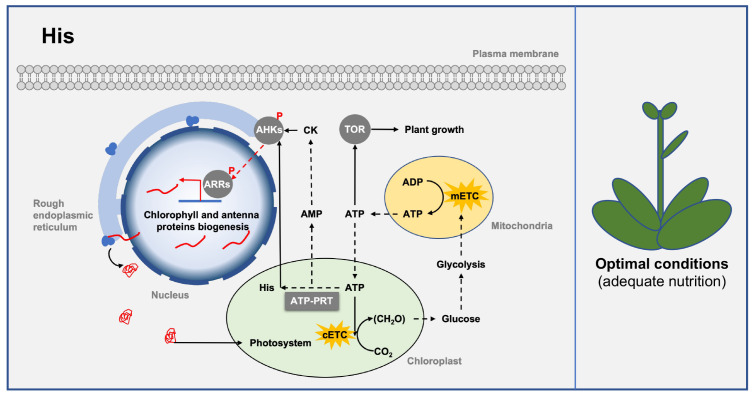
Class Ⅳ PRTase in the histidine biosynthesis pathway regulates plant growth and development. Class Ⅳ PRTase ATP–PRT regulates chloroplast development and plant growth under nutrient-adequate conditions by influencing intracellular His level and AMP/ATP metabolism. ATP–PRT is represented by a square box, and other proteins are represented by round boxes. Solid arrows indicate direct interactions or one-step reactions, and dashed arrows indicate indirect interactions, multistep reactions, or transmembrane transport. Abbreviations for proteins are as follows: Arabidopsis histidine kinase (AHK), Arabidopsis response regulator (ARR), target of rapamycin (TOR), ATP phosphoribosyltransferase (ATP–PRT). Abbreviations for metabolites are shown in the Abbreviations list.

**Figure 6 ijms-24-11828-f006:**
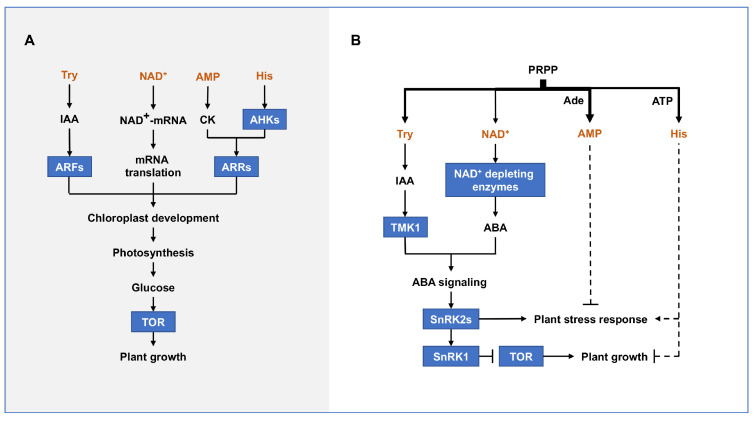
The PRTase-related metabolic pathways co-regulate plant growth (**A**) and abiotic stress response (**B**). Proteins are represented by square boxes. Dotted lines indicate unclear roles and mechanisms. Abbreviations for proteins are as follows: auxin response factor (ARF), Arabidopsis histidine kinase (AHK), Arabidopsis response regulator (ARR), target of rapamycin (TOR), transmembrane kinase 1 (TMK1), sucrose non-fermenting 1-related protein kinase 2 (SnRK2), sucrose non-fermenting 1-related protein kinase 1 (SnRK1). Abbreviations for metabolites are shown in the Abbreviations list.

**Figure 7 ijms-24-11828-f007:**
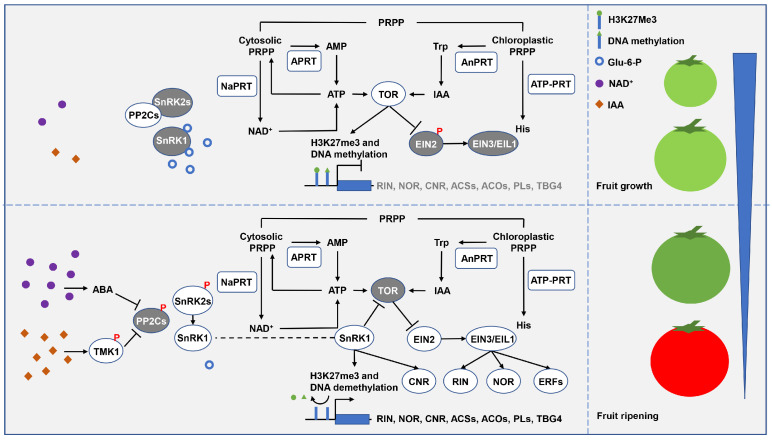
A possible model for the regulation of fruit growth and ripening by the PRTase-related metabolic pathways. Under nutrient-enriched conditions, SnRK1 activity is inhibited, and ATP and moderate levels of IAA synergistically activate TOR to promote fruit growth. However, under nutrient-limited conditions, high levels of NAD^+^ and IAA activate ABA signaling, so activated SnRK1 inhibits TOR activity and promotes fruit ripening. PRTases are represented by square boxes, other proteins are represented by round boxes. Active proteins are white, and inactivated proteins are grey. Abbreviations for proteins and genes are as follows: transmembrane kinase 1 (TMK1), protein phosphatase 2C (PP2C), sucrose non-fermenting 1-related protein kinase 2 (SnRK2), sucrose non-fermenting 1-related protein kinase 1 (SnRK1), nicotinamide phosphoribosyl transferase (NaPRT), adenine phosphoribosyltransferase (APRT), target of rapamycin (TOR), anthranilate phosphoribosyltransferase (AnPRT), ATP phosphoribosyltransferase (ATP–PRT), ethylene insensitive 2 (EIN2), ethylene-insensitive proteins (EIN3/EIL1), colorless non-ripening (CNR), ripening inhibitor (RIN), non-ripening (NOR), ethylene-responsive factor (ERF), ACC synthase (ACS), ACC oxidase (ACO), pectate lyases (PL), tomato beta-galactosidase 4 (TBG4). Abbreviations for metabolites are shown in the Abbreviations list.

**Table 1 ijms-24-11828-t001:** Classification of PRTases, their related metabolic pathways and experimental evidence of gene function.

Category	PRTase	Abbreviation	EC-Number	Related Metabolic Pathway	Transgenic Line	Phenotype	Refs
Class I	amidophosphoribosyltransferase	ATase	EC 2.4.2.14	purine nucleotide biosynthesis pathway	KO-*AtATase2*, KO-*NtATase2*	growth retardation and bleached/etiolated seedling	[8,11,15,34,35]
	adenine phosphoribosyltransferase	APRT	EC 2.4.2.7		KO-*AtAPRT1*, KO-*TaAPRT2*, KO-*OsAPRT2*, KO-*VaAPRT3*	delayed leaf senescence, gametophyte sterility, enhanced oxidation and high-temperature tolerance	[13,14,16,36,37]
	hypoxanthine-guanine phosphoribosyltransferase	HGPRT	EC 2.4.2.8		KO-*AtHGPT*	slowed seed germination	[38]
					OE-*AtHGPT*	accelerated seed germination	[38]
	orotate phosphoribosyltransferase	OPRT	EC 2.4.2.10	pyrimidine nucleotide biosynthesis pathway	—	—	—
	uracil phosphoribosyltransferase	UPRT	EC 2.4.2.9		KO-*AtUPRT*	light-dependent albino and dwarf	[10]
Class II	quinolinic acid phosphoribosyltransferase	QPRT	EC 2.4.2.19	NAD(P)^+^ biosynthesis pathway	KO-*AtQPRT*	embryo lethal	[9]
					KD-*AtQPRT1*	delay in growth and development	[39]
	nicotinamide phosphoribosyltransferase	NaPRT	EC 2.4.2.12		KO-*OsNaPRT1*	dwarfism and early leaf senescence	[12]
Class III	anthranilate phosphoribosyltransferase	AnPRT	EC 2.4.2.18	tryptophan biosynthesis pathway	—	—	—
Class IV	ATP phosphoribosyltransferase	ATP–PRT	EC 2.4.2.17	histidine biosynthesis pathway	OE-*AtATP–PRT* KO-*AtATP–PRT*	biomass reduction embryo lethal	[33] [40]

OE: overexpression; KD: knock down; KO: knock out.

## Data Availability

Not applicable.

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
