# Peer review of "Phosphoribosyltransferases and Their Roles in Plant Development and Abiotic Stress Response"

_ijms, 2023, doi:10.3390/ijms241411828_

Round 1
Reviewer 1 Report
The review, in my opinion meets the high stadards of the journal! Only minor corrections, listed below are needed.
I am not qualified to assess the quality of English in this paper. However, for me, the review is easy to read and quite undesrstandable!
General comment for the figures – it is not very clear whether they are compilation of already existing data (and if ”yes” there should be a respective citation) or they represent models developed and suggested by the authors.
The figures, if possible should be with higher resolution! In addition – the legends are very detailed and thus, too long. Since there is an Abbreviations list the names of the compounds could be referred referring to the list!
3.4. Class III PRTase should be 3.4. Class IV PRTase
Reviewer 2 Report
The manuscript could it be published as it stands. However, I would recommend the authors to carefully read again the manuscript and correct some mistakes reagarding the articles referenced. Some references do not match with what the authors report.
I am very glad the authors wrote this manuscript. It is a well-written, needed, and useful review . The authors are analytical and the tables/shcematics that they are presenting indeed help the author very much. I belive that the manuscript could be published prior to some minor revision. And after the authors have checked again some minor typos that exist.
minor corrections
Author Response
Please check the file enclosed.
